# *Crocus speciosus* (Iridaceae)—A New Species for the Bulgarian Flora

**DOI:** 10.3390/plants12040932

**Published:** 2023-02-17

**Authors:** Elena Apostolova-Kuzova, Kiril Stoyanov, Tsvetanka Raycheva, Samir Naimov

**Affiliations:** 1Department of Plant Physiology and Molecular Biology, University of Plovdiv, 4000 Plovdiv, Bulgaria; 2Department of Botany and Agrometeorology, Agricultural University, 4000 Plovdiv, Bulgaria

**Keywords:** *Crocus speciosus* subsp. *ibrahimii*, ITS region, morphology

## Abstract

This is the first report on the autumn-flowering species *Crocus speciosus*, belonging to *C*. ser. *Speciosi* from the Bulgarian flora. The species was found in Southeastern Bulgaria, in the area between Ahtopol and Rezovo. Re-examining the Bulgarian collections, the earliest specimen was collected in 1975, was probably overlooked, and most likely determined as *C. pulchellus*. The nearest known localities of the species are on the territory of Türkiye. In this study, we compared *C. pulchellus* and *C. ibrahimii* using DNA sequence data from the nuclear ITS1/2 region and morphological features. Our study showed a close relationship between the specimens from Bulgaria and the recently deposited data of *C. speciosus* and their separation from the closely related *C. pulchellus*. Together with the previously cited white anthers as a key feature for determination, the molecular data confirmed a clear distinction between the samples with white anthers in the two species. The morphological data of our taxon overlapped with the description of *C. ibrahimii*. The molecular data strongly supported the affiliation of *C. speciosus* s.l., but did not support the recognition of *C. ibrahimii* as a separate species and it should be referred to as a subspecies of *C. speciosus* subsp. *ibrahimii* Rukšāns.

## 1. Introduction

The genus *Crocus* L. includes approximately 200 species, which are divided into 2 subgenera—*Crocus* and *Crociris* B. Mathew [1]. The type subgenus contains the sections *Crocus* and *Nudiscapus* B. Mathew, containing 15 series. The main criteria of this taxonomy are based on morphological parameters. Many of the diagnostic markers used are quantitative, with no sharp boundary between parameters, e.g., the length of perigone segments, the coloration of perigone and anthers, etc. The representatives of this genus are widespread in Eurasia. Numerous variations in morphology are characteristic of populations, especially those from the Balkan Peninsula and Anatolia [1].

*Crocus speciosus* M. Bieb. and *C. pulchellus* Herb. are closely related, hysteranthous, autumn-flowering species. Both species are mesophytic plants occurring in S.E. Europe and Western Asia, mainly in the Balkan Peninsula and the Euro-Siberian phytogeographical region, extending to the Hyrcanian Province in Iran [2]. *Crocus speciosus* M. Bieb. is the type species of *C.*, ser. *Nudiscapus*, series *Speciosi* B.Mathew. Until recently, three taxa of the *Speciosi* series were known—the type subspecies from Georgia and two described by Brian Mathew from Türkiye—subsp. *ilgazensis* Mathew and subsp. *xantholaimos* Mathew [1]. Rukšāns later described subsp. *archibaldiorum* from Iran and subsp. *ibrahimii* from Türkiye in Europe, and recently, four more of this series from Greece and Türkiye [3]. This has been followed by *C. striatulus* Kerndorff & Pasche from N.E. Türkiye [4] and *C. brachyfilus* I. Schneider from Central Türkiye [5]. The use of molecular biological analyses applied to genus representatives has brought an entirely new understanding of crocuses, which defines the genus as complex and with species that are difficult to identify [4].

For the flora of Bulgaria, only two autumn-flowering species, *C. pulchellus* and *C. pallasii* Goldb., are known so far. Although we found a specimen deposited as *C. speciosus* in Bulgarian herbariums, the species has remained neglected. So far, all autumn-flowering specimens of the genus *Crocus* with white anthers in Bulgaria have been perceived as *C. pulchellus*. The region of Strandzha and the Black Sea coast has been characterized by peculiar ecological conditions, which is the reason for large morphological variations in some *Crocus* populations in the *C. speciosus* group. This has led to the unclear taxonomic status of some of the regional taxa and the elevation of subspecies to the species level based on morphological characters. Some of these species are not accepted in the World Checklist of Selected Plant Families [6] (e.g., *C. elegans* (Rukšāns) Rukšāns), nor have they been studied molecularly (such as *C. ibrahimii*). In this regard, we undertook a molecular study of populations of *Crocus* cf. *speciosus* with white anthers and unusually branched stigmas of an intense orange-red color, localized in a large continuous range of enlightened oak forests in the southern subregion region of the Black Sea coast, from Ahtopol to Rezovo in Bulgaria. To refine the phylogenetic position of these species, we sequenced the internal transcribed spacer region (ITS) which is one of the most used markers in phylogenetic research.

## 2. Results

### 2.1. Description Based on Bulgarian Materials

*Crocus speciosus* is an autumn hysteranthous geophyte (Figure 1). Its corm is oval-orbicular, with a height of 9.9–24 mm and a diameter of 10.4–22.6 mm, with separating basal rings that are 1.1–3.6 mm thick, and an irregular slight dentation. The corm neck is 3–4 mm long, with uneven loose fibers. It has tunic scales with a height of 5.6–12.8 mm. The plant’s height is 100–200 mm and prophyll is not present. It has 3–4 cataphylls that are silvery, with the lowest being 7.3–24 mm long and the highest 33.6–83.8 mm long. It has 2–3 leaves (4), which are glabrous to sparsely hairy. They are an intense green color and 2.5–4 mm wide. It has 1–2 flowers and a perigone tube with or without violet veining. The perigone throat is yellow and glabrous. The perigone segments are 22–56 mm × 6–27 mm, and are more often acuminate, and less often rounded. They are light to deep blue, on both sides, with more or less deep dark blue veins—the main 3–5 are visible and many are iridescent. The filaments are yellow (6–12 mm long), without indumentum or with sparse short hairs. The anthers are 7–22 mm long, with an arrow-shaped base, and they are creamy white, with colorless connective tissue. Their style extended above the stamens, repeatedly branching into numerous filiform stylodia, which are orange-reddish, with the length of the branched part being 10–24 mm. The capsules and seeds not observed.

A major key feature to distinguish *C. speciosus* from *C. pulchellus* is the branching and length of the stylodia (Figure 2) and their position compared to the anthers. Another good key feature is the indumentum of the filaments. Table 1 shows a comparison between the *C. pulchellus* and *C. speciosus* subsp. *ibrahimii* specimens from Türkiye (according to Rukšāns) and Bulgaria. Table A1 shows the measured features of the collected plants from Bulgaria.

### 2.2. Distribution

*Crocus speciosus* is widespread, but the specimens with white anthers have been observed only in the European part of Türkiye and Bulgaria (Figure 3). The known Bulgarian localities are from the south coast of the Black Sea, on the eastern foot of Strandja Mountain. Some observations from Türkiye from the higher part of the same mountain (Yıldız Dağları) suggest that the species could be found in the whole floristic region. Along with horological data for *C. speciosus* from Bulgaria, herbarium data for *C. pulchellus* from nearby locations are also found.

### 2.3. Chorology and Participation in Local Vegetation

The flowering specimens were observed from October to November. Leaves were noticed on collected and cultivated plants from January to February.

The specimens (Figure 4) were collected and observed in forests of *Quercus cerris* L., *Q. frainetto* Ten., *Q. pubescens* Willd. and *Carpinus orientalis* Mill. (Figure 5), with an undergrowth of *Cerastium dubium* (Bast.) Guepin, *Hippocrepis emerus* (L.) P. Lassen, *Juniperus deltoides* R.P. Adams, *Phillyrea latifolia* L., *Prunus spinosa* L., and *Pyrus nivalis* Jacq, and accompanying species of *Alliaria petiolata* (M. Bieb.) Cavara & Grande, *Arum italicum* Mill., *Briza maxima* L., *Cardamine pratensis* L., *Carex pendula* Huds., *Carduus candicans* Waldst. & Kit., *Centaurium erythraea* Rafn., *Clinopodium menthifolium* (Host) Stace, *Crocus adamioides* Kernd. & Pasche, *C. olivieri* J.Gay, *Galanthus nivalis* L., *Geranium sanguineum* L., *G. pyrenaicum* Burm. F., *G. robertianum* L., *Iris sintenisii* Janka., *Luzula forsteri* (Sm.) DC., *L. luzuloides* (Lam.) Dandy, *Medicago orbicularis* (L.) Bartal., *Melissa officinalis* L., *Mentha aquatica* L., *M. pulegium* L., *Mercurialis ovata* Sternb. & Hoppe, *Plantago major* L., *Polygala supina* Schreb.; *Prospero autumnale* (L.) Speta, *Prunella vulgaris* L., *Rumex patientia* L., *R. pulcher* L., *Ruscus aculeatus* L., *Salvia virgata* Jacq., *Symphytum tuberosum* L., and *Trisetum flavescens* (L.) P. Beauv. Some specimens were observed on the open land over the sea coast, in shrubs of *Phillyrea latifolia* and *Pyrus bulgarica* Khutath. & Sachok., with accompanying *Asparagus acutifolius* L., *Carthamus lanatus* L., *Cistus creticus* L., *Colchicum autumnale* L., *Eryngium maritimum* L., *Ficus carica* L., *Galatella linosyris* (L.) Rchb. F., *Gastridium ventricosum* (Gouan) Schinz & Thell.; *Ornithogalum sigmoideum* Freyn & Sint., *O. umbellatum* L., *Osyris alba* L., *Trifolium cherleri* L., and *Verbascum phoeniceum* L.

The representatives of *C. speciosus* have occurred in the territory of Strandzha Nature Park, and Natura2000 site BG0001007. The specimens SOA 063308 and 063309 were collected on the border of the “Oustie na reka Veleka” protected site. The observation area of SOA 063310 and 063311 was substantially inside the “Silistar” protected site and along its western border. The specimen of SOA 063312 was on the border of the same site in a place intensively used as an RV camping area. The collection of SOA 035125 was not confirmed because of the insufficiently described location without coordinates. The landscape of the supposed locality has changed during the last 45 years—the place is an artificial pine forest, with a high level of ruderalization, and is often used as a picnic area and for illegal camping. All occurrences were from the floristic region of the Black Sea Coast (southern) but, according to the data from Türkiye, it is very likely to be found in the internal area of Strandja Mt. floristic region.

### 2.4. ITS Sequences

*Crocus speciosus* and *C. pallidus* were distinguished using the sequences of the nuclear rDNA ITS1/2 region (Figure A1), as the samples of the two species showed minor differences in the margins of each species. The obtained sequences were not interrupted and had no unidentified bases, from which it follows that they do not have different variants in the genomic DNA of the studied samples. The differentiation of all *C. pulchellus* samples from *C. speciosus* was clear, e.g., it could be illustrated with the insertion of CCTCC on position 160 (Figure A1). The constructed phylogenetic tree followed the known topology [7], with the two species separated into their own branches, in the node of series *Speciosi*. At the same time, there was no arrangement in *C. speciosus* to separate subsp. *ibrahimii* from the other infraspecific taxa—subsp. *speciosus*, subsp. *ilgazensis*, and subsp. *xantholaimos* (Figure 6).

## 3. Discussion

In most of the determination keys, white or creamy white anthers have been indicated only for the autumn-flowering *Crocus pulchellus*, which is why a large number of taxa with white anthers from the region of Türkiye and Bulgaria have long been designated as this species. Of the so far described taxa of the *C*. ser. *Speciosi*—*Crocus speciosus* complex, the white anthers have been indicated for *C. elegans*. It was originally described as a subspecies of *C. speciosus* subsp. *elegans* Rukšāns, [8]. Subsequently [9], fluctuations in this character were noted—when visiting the population in the next growing season, the anthers turned out to be yellow. In the same period, Schneider [5] described *C. brachyfilus* from the Mediterranean region of Konya in Turkey. Both creamy white and yellow anthers were reported for the species. Distinctive features remained, such as thin corm rings and an extremely developed corm neck, white filaments, and yellow, rarely creamy white, anthers.

This unusual occurrence of white anthers in the *Crocus speciosus* complex was found in Türkiye by Rukšāns [3]. A taxonomic rank was given to the subspecies *C. speciosus* ssp. *ibrachimii* and later was elevated to the species rank *C. ibrahimii* (Rukšāns) Rukšāns, [9]. This concept was based on morphological and geographical parameters but not by using molecular evidence. The range of this taxon is about 60 km west of Istanbul, near the Bulgarian border, and, according to Rukšāns, it would not be surprising if it also occurred in southeastern Bulgaria [9,10].

Sequences of the ITS1/2 region showed a clear grouping of specimens by species, but a lack of a more detailed arrangement within the boundaries of the species, regardless of their anther coloration (Figure 5). This leads us to accept the original subspecies rank of *C. s.* subsp. *ibrahimii*.

So far, no detailed studies have been carried out on crocuses in Bulgaria, which is why the species remained unnoticed due to its white anthers, probably being considered *Crocus pulchellus*. In the Herbarium of Agricultural University—Plovdiv (SOA), a single preserved specimen was found, identified as *C. speciosus*, from the region of Akhtopol (Delipavlov, SOA 035125), which has not been taken into account when summarizing the species diversity of crocuses in our country and has not yet been included in floristic publications.

At first sight, our collected samples are morphologically close to *C. pulchellus* and *C. ibrahimii* when compared to the other taxa from ser. *Speciosi*. The analysis of the morphological characters showed a close similarity to *C. ibrahimii* (Table 1). The perigonal segments in *C. ibrahimii* and *C. speciosus* are larger (longer than 35 mm), compared to those of *C. pulchellus* (average up to 33 mm). The anthers’ color and size did not show discrete differences between the taxa. However, the most significant differences are the multiply divided stylodia, exceeding the height of the anthers, as well as the intensive orange coloring, compared to the pale coloring of the stigma in *C. pulchellus*. In addition, compared to *C. pulchellus*, filament papillae are absent in *C. ibrahimii*, as described by Rukšāns and observed in our collected samples. Both species, compared in the spring, show morphological differences in the leaves. There are 2–3 leaves of *C. speciosus* subsp. *ibrahimii*, exceptionally four, while in *C. pulchellus,* there are 4–5. Additionally, the leaves of the observed plants of *C. speciosus* subsp. *ibrahimii* are not wider than 4 mm, while the leaves of the observed specimens of *C. pulchellus* are 4–5 mm wide at the same time.

The analysis of the phylogenetic tree in our study showed that the specimens from Bulgaria and the included deposited data of *C. speciosus* have a close relationship and are separate from the closely related *C. pulchellus*. The specimens of *C. speciosus* remain unconstrained, together with the known *C. speciosus* subsp. *xantolaimos* and subsp. *ilgazensis* (Figure 5—HE801120, HE801124). Both species are united in the node of ser. *Speciosi*. The sequences of the ITS1/2 region showed a clear grouping of the samples by species, but a lack of a more detailed arrangement within the species boundaries, regardless of their anther coloration. This leads us to accept the original subspecies rank of *C. s.* subsp. *ibrahimii*.

The number of ITS1/2 rDNA sequences of *C. speciosus* deposited in the NCBI is limited and there are no records for *C. ibrahimii*. However, the clustering of our samples confirms a genetic proximity and close relationship with *C. speciosus* from Turkey (Figure 5, Table A2). The taxonomic relationships within the *Crocus speciosus* complex have not yet been fully explored. Therefore, it is not clear whether the taxa of the complex have the status of species or subspecies. Despite earlier proposals to raise *C. ibrahimii* to the species level [9], our study displayed low levels of genetic divergence. Therefore, returning the taxon to the subspecies level is correct.

The morphological variation of ambiguous characters in *C. speciosus* s.l., together with the molecular results indicate the genetic homogeneity of *C. speciosus* and uncertainty for the taxonomic treatment of *C. ibrahimii* at the species rank. Therefore, we consider *C. ibrahimii* (Rukšāns) Rukšāns to be a synonym and its downgrade in rank is justified. It is also the earlier legitimate name of the taxon [8] that has priority.

## 4. Materials and Methods

### 4.1. Examined Specimens

The samples of *C. speciosus* and *C. pulchellus* were collected during terrain work in the period 2019–2022. The examined specimens are listed below. New and unpublished chorological data are signed with an asterisk (*). The vouchers of the collected specimens were deposited in the Herbarium of Agricultural University—Plovdiv (SOA). The other revised specimens were from the herbaria SOA, SOM—Institute of Biodiversity and Ecosystem Research at the Bulgarian Academy of Sciences, and SO—Faculty of Biology at Sofia University. The data about the type specimens of *C. ibrahimii* from Herbarium GB, University of Gothenburg, were taken via GBIF.org [11]. Some observations of points on the map are cited from iNaturalist.org. The specimens are listed by country, floristic region (in bold), MGRS square (10×10 km, in bold), description, decimal geographic coordinates (WGS84, if present), elevation, date, and collector, followed by herbarium acronym and entry number. The voucher specimens for the examined ITS sequences are signed with their NCBI GenBank numbers in brackets. The morphologically measured specimens are signed with “(M)”. 

*Crocus speciosus* subsp. *ibrahimii*:

BULGARIA: *Black Sea Coast (southern): 35TNG76: Grasslands North of Akhtopol town, 6 October 1977 (coll. D.Delipavlov) SOA 035125; 35TNG84: Oak forest NW of Rezovo Village, N41.99428 E28.02181, 63 m, 21 October 2022 (coll. T.Raycheva and K.Stoyanov) SOA 063311 (M); 35TNG85: “Sekana Koriya” locality—the left coast of Veleka River near Sinemorets Town, N42.06027 E27.97088, 37 m, 21 October 2022 (coll. T.Raycheva and K.Stoyanov) SOA 063308 (M, OQ096622); N42.06009 E27.96753, 26 m, 21 October 2022 (coll. T.Raycheva and K.Stoyanov) SOA 063309 (M, OQ096621); “Golemiyat pyasak” locality and “Klencheto” peak on the path to “Lipite” beach, N42.04138 E27.99132, 86 m, 21 October 2022 (coll. T.Raycheva and K.Stoyanov) SOA 063310 (M); the estuary of “Silistar” river, N42.02362 E28.00757, 4 m, 21 October 2022 (coll. T.Raycheva and K.Stoyanov) SOA 063312 (M, OQ096620); TÜRKIYE: 35TPF26: Türkiye, Yildiz Daglari, near Çanakça, N41.2568889 E28.4893083, alt. 100 m, ex culturae in Horto Ibrahim Sozen 12 October 2008 (I.Sozen/Rukšāns) GB-0152377!—Holotypus, GB-0105890!—Isotypus [11]; 35TNG34: Dereköy, N41.9322618 E27.3656639, 13 November 2020 (Furkan Kilic, georeferenced photo https://www.inaturalist.org/observations/141914961 accessed on 20 December 2022); 35TNG43, Demirköy, N41.8474917 E27.527078, 18 October 2020 (Engin Asav, georeferenced photo https://www.inaturalist.org/observations/63312682 accessed 20 December 2022 )

*Crocus speciosus* subsp. *speciosus*

GEORGIA: 37TFH37: Ouathara River, alt. 1000 m (coll. A. Kolakovskiy) 14 September 1936 SOM 13859; N. MACEDONIA: 34TEM82: Delisinci, 20 October 1967 (coll. K. Micevski) SOM 157734; UKRAINE: Crimea: 36TXQ03: Nikitskaya Yayla, 7 October 1959 SOA 04272, 04277; 36TWQ82: Ai-Petri, 20 March 1901 SOM 13863; (coll. B. Kitanov and L. Pivalova) 19 September 1958 SO 32105. 


*Crocus pulchellus*


BULGARIA: Sredna-Gora Mts. (western): South of “Boev Shamak” peak, N42.562 E24.331, 375 m, 2020-11-10 (coll. T.Raycheva and K.Stoyanov) SOA 062992 (M); Rhodopi Mts. (central): 35TLG53: “Velichka” river, under the rock niches “Dikilitash”, near Boyan Botevo village; N41.846667 E25.272222, N41.857222 E25.243056, 350 m, 16 July 2020 (coll. V. Trifonov) SOA 062985 (M); Rhodopi Mts. (eastern): 35TLF57: Kosturino village, Kirkovo Municipality, N41.3443333 E25.2935, 571 m, 14 October 2020 (coll. V. Trifonov) SOA 062986; 35TMF28: under “Kodjakaya” peak, near Meden buk village, N41.44983 E26.13808, 220 m, 21 October 2020 (coll. V. Trifonov) SOA 062990; 35TMG11: under “Kush Kaya” peak, N41.71047 E26.00875, 375 m, 21 October 2020 (coll. V. Trifonov) SOA 062991; 35TMG12: The path to “Gluhite Kamani”, near Ivaylovgrad town, N41.7317 E25.9689, 554 m, 14 November 2019 (coll. T.Raycheva and K.Stoyanov) SOA 062842 (OQ096618); Tracian lowland: 35TLG85: Oak forest near Brod village, N42.05697 E25.66899, 127, 9 October 2019 (coll. T.Raycheva and K.Stoyanov) SOA 063313 (OQ096619).

The collected and observed samples were mapped using QGIS [12], over a layer from OpenStreetMap.org. The data for the accompanying species were taken by determination of the specimens in the locality, and from herbarium data with exact coordinates.

The collected plants from each locality were measured for each described feature using a caliper. The measurements were processed using basic statistical analysis, providing the minimum, maximum, average and standard deviation, as shown in Table 1 and Table A1. 

### 4.2. DNA Extraction, Amplification and Sequencing

Plant material was immediately put in liquid nitrogen and stored in a laboratory at −80 °C before starting the molecular analysis. The plant genomic DNA was purified as described using a DNeasy Plant Mini Kit (QIAGEN, Hilden, Germany) according to the manufacturer’s requirements as described earlier [13]

The quality of the resulting DNA was assessed spectrophotometrically, using an Epoch™ Microplate Spectrophotometer, USA, and DNA integrity was evaluated by 1% agarose gel electrophoresis.

The DNA fragment encoding for the ITS1—5.8S rDNA—ITS2 cluster was amplified using the following primers: ITS-A (5′–GGAAGGA-GAAGTCGTAACAAGG–3′) and ITS-B (5′–CTTTTCCTCCGCTTATTGATATG–3′) [14], as described earlier [15]. The reactions, set in a final volume of 50 μL contained 1x reaction buffer, 200 μM of dNTPs, 0.2 μM of each primer, 100 ng of genomic DNA, and one unit of Q5 High Fidelity DNA polymerase (New England Biolabs). The PCR amplification was conducted under the following parameters: initial denaturation at 94 °C for 45 s, followed by 30 cycles at 94 °C for 10 s for denaturation, 10 s at 62 °C for primer annealing, 30 s at 72 °C for primer extension, and a final elongation step of 2 min at 72 °C. Amplified PCR products were separated by 0.8% agarose gel electrophoresis, excised from the gel, and purified using a QIAquick Gel Extraction Kit (QIAGEN). The purified DNA fragments were bidirectionally sequenced in Eurofins facility.

### 4.3. Phylogenetic Analysis

The obtained nucleotide sequences were blasted against those from the NCBI Nucleotide database [16,17]. This bioinformatic analysis proved the origin of our nucleotide sequences. Newly obtained sequences were deposited in the NCBI database and are accessible through the corresponding accession numbers OQ096618—OQ096622 (Figure 6, Table A2). The best hits were downloaded and used for phylogenetic analysis. The alignment of the sequences was achieved using the ClustalW Multiple alignment software [18]. The phylogenetic analysis was conducted using Bayesian phylogenetic inference with MrBayes 3.2 software [19]. The parameters of the analysis were the same as described by Harpke et al. [20]: 2 × 4 chains for two million generations, nuclear data set ГTP + G + I, sampling tree per 1000 generations, and two independent runs. The result was visualized as a phylogenetic tree using TreeGraph 2 [21]. The analysis included 31 nucleotide sequences, cited as numbers of GenBank entries in the phylogenetic tree (Figure 6, Table A2). Because we used the analysis to find the place of the evaluated taxon, we compared the result with the known phylogeny [7,20]. For this task, we used taxa from the series *Speciosi*, *Biflori* and *Reticulati* to find the phylogenetic position of the specimens from Bulgaria. We accepted *C. nudiflorus* and series *Laevigatae* for the outgroup. The result of the alignment was visualized in Figure A1 using CLC Sequence Viewer [22].

## 5. Conclusions

There are no deposited ITS1/2 sequences of *Crocus speciosus* subsp. *ibrahimii* Rukšāns in the NCBI Nucleotide database. The morphological features of the collected and observed samples in this study are very close to the known morphology of this taxon, and closer than the so far known taxa in *C*. ser. *Speciosi*. The phylogenetic analysis strongly confirms the affiliation to *C. speciosus* s.l. and does not support the recognition of this taxon as a different species. According to the above, this taxon should be classified as a subspecies.

Despite the similarity in the white anthers, *C. speciosus* could easily be distinguished by morphology from *C. pulchellus*. Certain key characteristics are the shape, length and coloration of the stylodia, the indumentum of the filaments and the width of the leaves.

The geographic location indicates an expansion of the range of the taxon in the Balkans. The white anthers in almost all identification keys in the region lead directly to *C. pulchellus*. It is possible that *C. speciosus* subsp. *ibrahimii* could also be found in other areas of the Balkan Peninsula.

## Figures and Tables

**Figure 1 plants-12-00932-f001:**
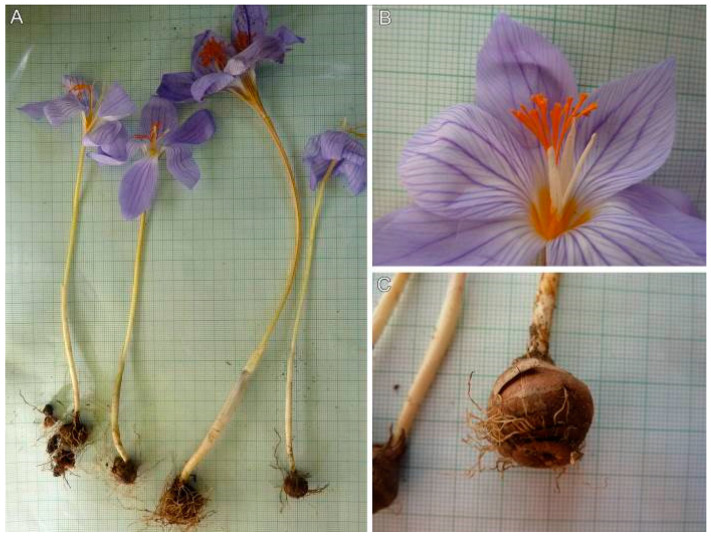
Morphological features of *Crocus speciosus* (SOA 063310, grid 1 mm): (**A**) whole plant; (**B**) anthers and stylodia; (**C**) corm.

**Figure 2 plants-12-00932-f002:**
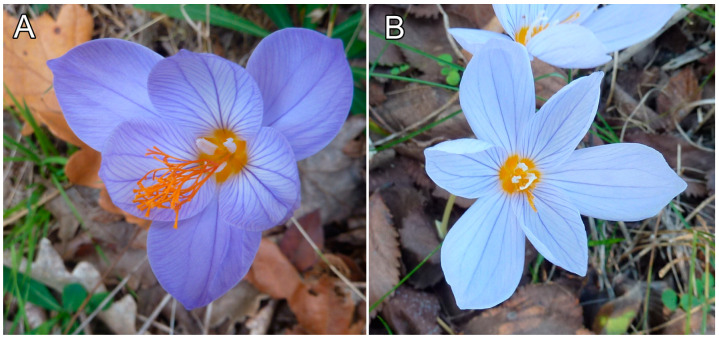
Flower: (**A**) *Crocus speciosus*; (**B**) *C. pulchellus*.

**Figure 3 plants-12-00932-f003:**
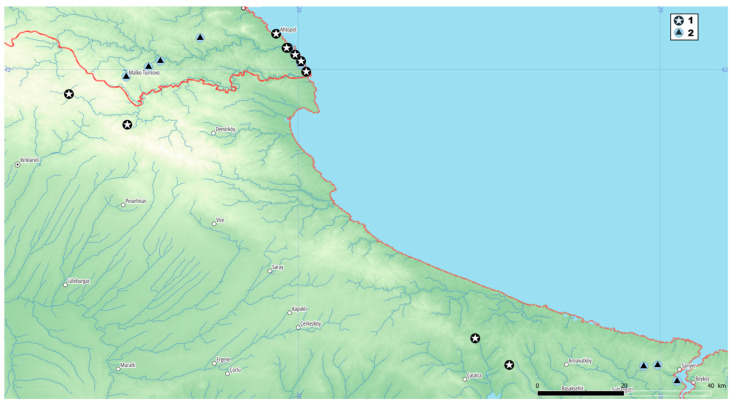
Occurrences in Bulgaria and Türkiye: (1) *Crocus speciosus subsp. ibrahimii*; (2) C. *pulchellus*.

**Figure 4 plants-12-00932-f004:**
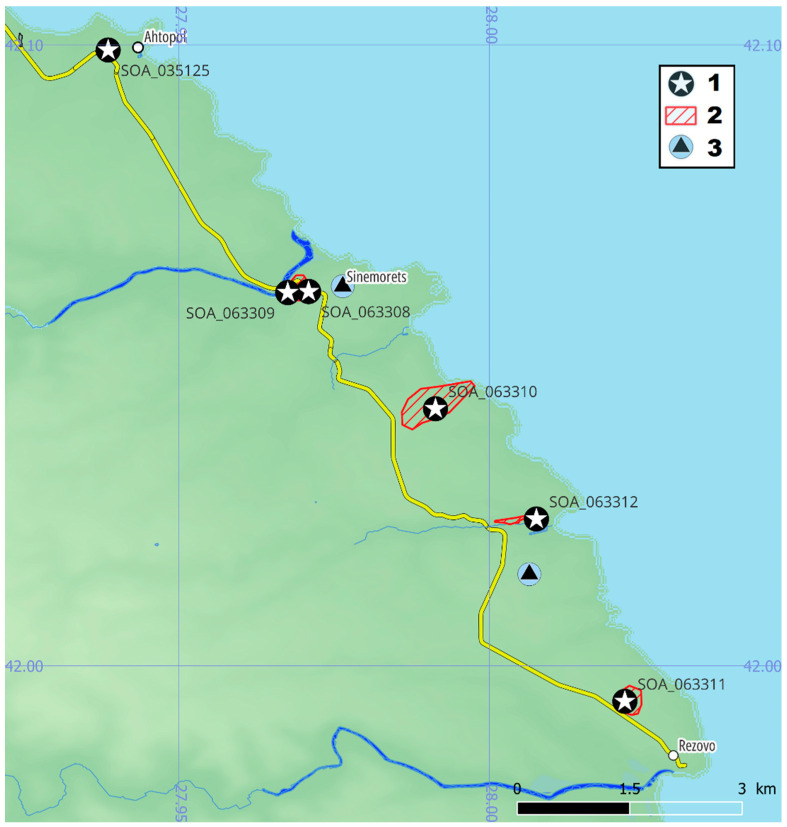
Specimens of *C. speciosus* subsp. *ibrahimii* from Bulgaria: (1) collected specimens; (2) observed specimens; (3) data entries of *C. pulchellus*.

**Figure 5 plants-12-00932-f005:**
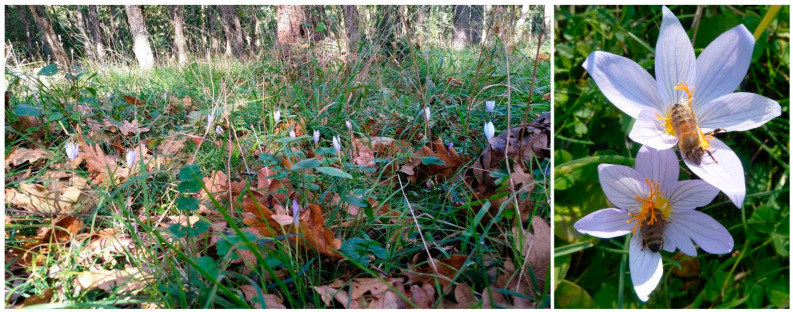
Habitat and pollination of *C. speciosus*.

**Figure 6 plants-12-00932-f006:**
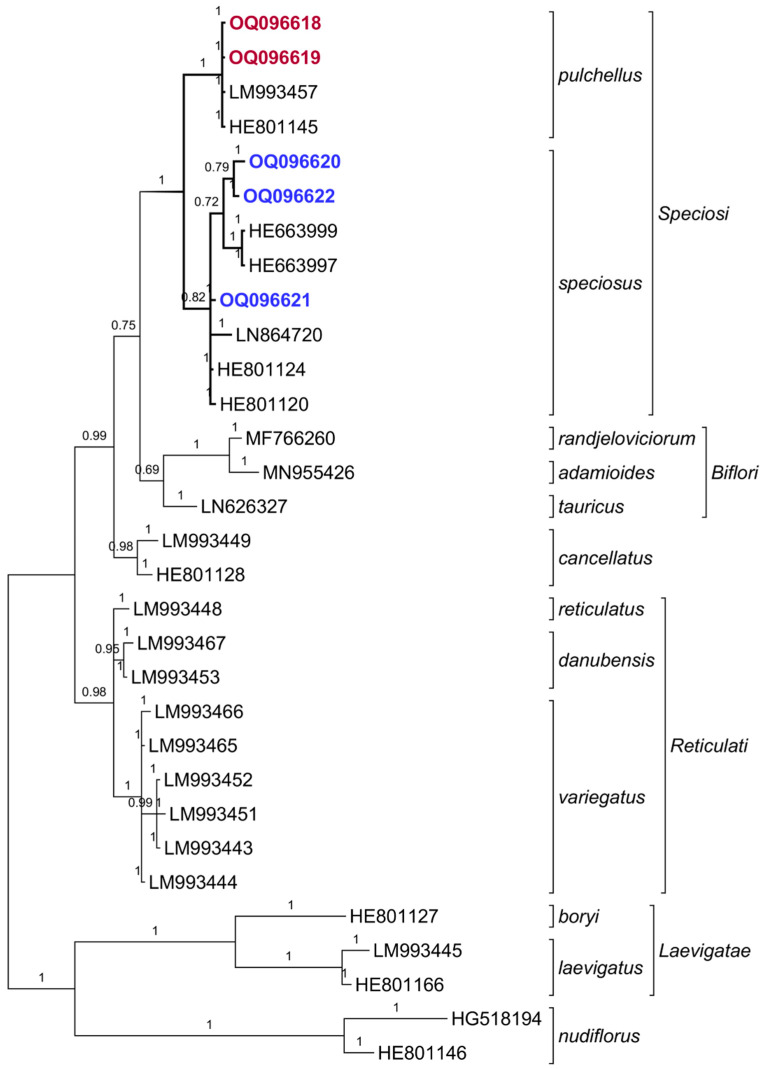
Placement of *Crocus speciosus* in the phylogenetic tree of sect. *Nudiscapus* obtained from a Bayesian phylogenetic inference of the nuclear rDNA ITS regions using the methodology of Harpke et al. [7]. Posterior probabilities are designated by numbers. See Table A2 for details.

**Table 1 plants-12-00932-t001:** A comparison of the morphological features of *Crocus pulchellus* and *C. speciosus* subsp. *Ibrahimiii **.

Character	*C. pulchellus* (Current Data)	*C. speciosus* subsp. *ibrahimii* (Current Data)	*C. ibrahimii* (according to Rukšāns)
Leaves, count	4–5	2–3 (4)	3–4
Leaf width, mm	4–5	2.5–4	2.5–4
Perigone tube	Deep yellow	Yellow	Yellow
Perigone segments, length, mm	21.8–52.1 33.38 ± 4.45	22.2–56 38.06 ± 5.34	35–38–40
Perigone segments, width, mm	7.1–18.6 12.78 ± 2.05	6.3–26.9 14.7 ± 2.9	8–13–15
Filaments, length, mm	4.7–12.1 8.16 ± 1.24	3.5–13.9 9.12 ± 2.1	6–10
Filaments, color	Yellow	Yellow	Yellow
Filaments, indumentum	hirsute	glabrous to glabrescent	glabrous to glabrescent
Anthers, length, mm	5.8–15 10.5 ± 2.05	7.3–21.9 13.8 ± 2.6	8–13.5
Anthers, color	White	White	White
Length of the branched stylodia, mm	6.3–17.3 13.3 ± 2.47	9.7–24 16.19 ± 3.03	10–14
Branching of the stylodia	Starting under the level of the anthers, three lobes, sometimes divided into three more	Starting on the level of the anthers, tree-like, multiple	Starting on the level of the anthers, tree-like, multiple
Color of stylodia	Pale orange to yellow	Deep orange	Deep orange

* See Table A1 for detailed measurements of the specimens.

## Data Availability

Data is contained within the article.

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
