# Peer review of "Crocus speciosus (Iridaceae)—A New Species for the Bulgarian Flora"

_plants, 2023, doi:10.3390/plants12040932_

Round 1
Reviewer 1 Report
Dear Authors,
The premises of your work are interesting and the paper is well presented (talking about style and figures). But I have some great scientific concerns. You well present morphological and other descriptive data in your work, but the moelcular part, which should be in my opinion the most consisten one, is very unconsistent. I believe that is not sufficient to analyse only one nucleotide sequence to affirm that you find a novel subsp. Even more, I believe you need to analyse the DNA of hundreds of samples, but in the "materials and methods" section I can see that you list the name of the samples but not how much samples you analysed. My main and big concern resides in the number of loci you need to analyse in order to correctly affirm that you find a novel subsp. Only the ITS is not sufficient.
I strongly recommend to improve the work, because it is quite interesting and well written, but by now I have to reject it, sine Plants is a Q1 journal and I believe you need to improve your molecular analysis in order to publish on a Q1 journal.
Author Response
Thanks for the review. Our study refers to the taxa discovered in Bulgaria. We agree that many molecular markers of samples from different regions and species are necessary to make a phylogenetic decision. Such is the long-term work of D. Harpke, the result of which we use as a basis to prove our viewpoint. In our case, the few samples compared with those deposited in NCBI are pretty sufficient to prove the taxon in the established range.
Reviewer 2 Report
The manuscript by Apostolova-Kuzova et al. describes the investigations both using molecular methods (ITS) and morphology of a closely related group of Crocus in the section Speciosi. Insofar as methodology and approach are considered, the study is sound and the results, although the overall sampling in the field seems not to have been very large, appear justified.
I would have liked to see descriptions of capsules and seeds, especially since floral traits have been studied extensively, the fruit state of the plant would have been worth a further observation. Was it not possible or did the authors not find a suitable spot to transplant the species in order to observe fructification, e.g., in a garden?
I believe the communication may prove worthwhile both for practitioners in the field and conservationists. A more thorough study of adjacent areas to get a broader geographical database would have been desirable.
Here come a few remarks to be handled that mainly concern the English and the spelling of botanical names:
Language / botanical names
1. Line 22 hysteranthous
2. Line 9 ...from the Bulgarian flora... // Line 13 is „despite“ the correct term? Shouldn‘t it be „together with...“?
3. Line 51 stringency of the use of hyphen (-) and em-dash (–): 100-200 mm (if so, e.g. in line 50 3-4 mm), same to be noticed in table 1. Please refer to the journal‘s style in this regard.
4. Line 82 abbreviation for Quercus is Qu.
5. Lne 83/84 Phillyrea not Phyllirea dto. line 91
6. Line 104 … it is very likely to be found... (replace)
7. Line 122 comma after the plant name (before „which)
8. Line 134/135 sentence is corrupted, probably add „and“ after ssp. ibrahimii to connect the two expressions in the sentence.
9. Line 137 … but not by using molecular...
10. Line 167 Crocus xantholaimis ibd. Crocus ilgazensis …
11. Line 177 Crocus ibrahimii
12. Line 289/290 italicize Latin plant names
Overall, I suggest getting the ms checked by a native speaker.
Author Response
Thanks for the great review. Of course, we have a living collection in which we are growing specimens collected from Bulgaria. We are working on a comparative study of the Bulgarian representatives of the genus Crocus on the morphology of the capsules and seeds.
Thank you for your careful reading and comments on the text. We agree with your suggestions.
Reviewer 3 Report
Dear authors
This is a very well written, easy to understand, and well suported paper.
I only have a few minor comments
Line 19: please avoid using as keywords those that are already mentioned in your title (Bulgarian flora)
Line 76: and Bulgaria (Figure 3).
Author Response
Thank you for the review. We agree with your comments.
Reviewer 4 Report
I read the manuscript with much interest because it shows the importance of using modern molecular methods in solving taxonomic problems. I have no fundamental comments regarding the technical, methodological, and statistical analyses use.
However, the section 2.3. Phenology and habitats is incorrectly titled. In the field of phenology, only the flowering period (October-November) was given, without any detailed analysis (i.e. the relationship between the plant development stage and habitat conditions as climate and site factors). As in ecology a habitat is defined as a place where an organism lives. A habitat meets all the environmental conditions an organism needs to survive. For a plant, a habitat must provide proper combination of light, air, water, and soil. That why this term refers directly to abiotic conditions. While in current work the authors provided the species composition of the vegetation accompanying crocus and the geographical distribution of the species' sites. Maybe the title “chorology and participation in local vegetation” (or similar) would be more appropriate.
Minor notes (text correction) are included in the attached file.

Author Response
Thank you for the remarks and suggestions. We agree with all of them.
Round 2
Reviewer 1 Report
As I specified during the first round of revision, in my opinion is not sufficient to analyse only one DNA sequence. In any case, I can see the other reviewers suggested the acceptance of the manuscript after minor revision. So, I maybe is better to follow their suggestion. Even because the authors did not reply to my comments, they just said that "it is sufficient to prove the taxon".